# Textile Bandwidth-Enhanced Polarization-Reconfigurable Half-Mode Substrate-Integrated Cavity Antenna

**DOI:** 10.3390/mi14050934

**Published:** 2023-04-25

**Authors:** Feng-Xue Liu, Jie Cui, Fan-Yu Meng, Tian-Yu Jiang, Shao-Fei Yan, Shuai Chao, Lei Zhao

**Affiliations:** 1Jiangsu Normal University Kewen College, Xuzhou 221132, China; 2School of Physics and Electronic Engineering, Jiangsu Normal University, Xuzhou 221116, China; 3Jiangsu Xiyi Advanced Materials Research Institute of Industrial Technology, Xuzhou 221400, China; 4School of Transportation Engineering, Jiangsu Vocational Institute of Architectural Technology, Xuzhou 221116, China; 5School of Information and Control Engineering, China University of Mining and Technology, Xuzhou 221116, China; leizhao@cumt.edu.cn

**Keywords:** textile antenna, wearable antenna, half-mode substrate-integrated cavity antenna, bandwidth enhancement, polarization reconfiguration

## Abstract

A textile bandwidth-enhanced polarization-reconfigurable half-mode substrate-integrated cavity antenna was designed for wearable applications. A slot was cut out from the patch of a basic textile HMSIC antenna to excite two close resonances to form a wide −10 dB impedance band. The simulated axial ratio curve indicates the linear and circular polarization of the antenna radiation at different frequencies. Based on that, two sets of snap buttons were added at the radiation aperture to shift the −10 dB band. Therefore, a larger frequency range can be flexibly covered, and the polarization can be reconfigured at a fixed frequency by switching the state of snap buttons. According to the measured results on a fabricated prototype, the −10 dB impedance band of the proposed antenna can be reconfigured to cover 2.29~2.63 GHz (13.9% fractional bandwidth), and the circular/linear polarization radiation can be observed at 2.42 GHz with buttons OFF/ON. Additionally, simulations and measurements were carried out to validate the design and to study the effects of human body and bending conditions on the antenna performance.

## 1. Introduction

In the last decade, significant progresses were made in the applications of wearable antennas in medical monitoring [1], sports [2], defense [3], security [4] and other areas. Among the mainstream wearable antennas, the textile antenna became a hot research topic because of its high portability, flexibility and integrability in textile materials [5,6,7,8,9]. The half-mode substrate-integrated cavity (HMSIC) antenna, and half-mode substrate-integrated waveguide (HMSIW) [10,11], is among the promising candidates as textile antennas for its flat geometry and halved 2-D footprint compared to the complete resonance cavity. The vertical sidewall defining the half cavity can be formed by uniformly arranged metallic rivets [12] or embroidered conductive threads [13,14]. However, due to the highly frequency-selective cavity geometry, the HMSIC antenna is of a small bandwidth which is not conducive to its reuse in multiple frequency bands. According to the literature, adding shorting vias or slot can increase the bandwidth for wearable/flexible HMSIC antennas. Reference [15] presented a cork-substrate slotted HMSIW antenna with two close hybrid modes to obtain a 23.7% fractional bandwidth around 5.5 GHz. In reference [16], a slot was added in a textile dual-mode HMSIC antenna to increase its fractional bandwidth from 4.5% to 6% at its lower 2.45 GHz band. Reference [17] introduced an all-textile PIFA with shorting pins and slot to manipulate its dual resonances of the TM_0,1/2_ and TM_0,3/2_ modes to obtain an enhanced 18% bandwidth around 5.5 GHz. Additionally, our previous works [17,18], respectively, proposed two bandwidth-enhancing methods by adding four shorting vias or a V-slot. The two designed textile HMSIC antennas showed similar enhanced bandwidths (14.7% for [17] and 15.7% for [18]) around 5.5 GHz, but the latter yielded a much smaller 2-D footprint which is favorable in wearable applications.

On the other hand, the reconfigurable antenna became one of the hottest research topics for wearable antennas [19]. The wearer can be in various positions and gestures, and the reconfigurable polarization enables the wearable antenna to achieve a flexible signal transmission. References [20,21] introduced two wearable antennas using PIN-diode-based DC bias circuits to electrically reconfigure the polarization modes. However, such bias circuits usually require soldering which could be difficult on textile materials and vulnerable against physical deformations and need extra DC powers which conflict with the limited battery capacities of wearable devices. References [22,23] introduced an improved solution for the bias circuit of the frequency-reconfigurable textile patch antenna by integrating the bias circuit into snap buttons, but DC powers are still required. In addition, reference [24] introduced a mechanical polarization reconfiguration method for the textile patch antenna by using conductive Velcro tape and zip fastener. Reference [25] also presented a textile coupled-mode substrate-integrated cavity antenna with mechanically reconfigurable patterns using snap buttons. As commonly used clothing accessories, the snap button, Velcro tape and zip fastener are highly compatible with most textile materials and require no DC powers to function as switches.

This paper presents a textile HMSIC antenna with an enhanced bandwidth and reconfigurable polarization for wearable applications. Firstly, a textile HMSIC antenna is designed and simulated to investigate its bandwidth. Secondly, a slot is added to enhance its bandwidth to cover the medical body area network (MBAN) and 2.45 GHz industry science medical (ISM) bands, and the linear polarization (LP) and circular polarization (CP) radiations are predicted at different frequencies based on the simulated axial ratio (AR) curve. Thirdly, two sets of metallic snap buttons are installed along the edge of the patch as switches to shift the −10 dB band, and simulations with buttons OFF and ON are carried out to illustrate its band and polarization reconfiguration characteristics. The frequency and radiation characteristics of the proposed antenna on the human tissues are simulated and analyzed, and the specific absorption rate (SAR) in the human tissues are, respectively, simulated and analyzed. Lastly, a fabricated prototype of the proposed antenna is measured in an anechoic chamber to validate the design, and the influence of bending conditions is measured and analyzed.

## 2. Basic Textile HMSIC Antenna

As shown in Figure 1, a basic textile HMSIC antenna (Antenna I) was designed. It consisted of a sheet of low-loss foam as the substrate, two sheets of conductive fabric as the patch and the ground, embroidered conductive threads as the sidewall and a coaxial probe as the feeding. The patch and ground were of a sheet resistance of 0.04 Ω/square. The selected substrate material was of a relative permittivity of 1.06 and a dielectric loss tangent of 0.0001, and the thickness of the substrate was 3.2 mm. The sidewall was of a measured sheet resistance of 0.6 Ω/square. The material of the feeding probe was copper, and its radius was set to be 0.6 mm to imitate the SMA connector. The ground and the substrate were selected to be of a same square shape with a side length of 120 mm to shield the radiation toward the human body. The radiation aperture was located along the center line of the top surface of the substrate, and the patch was, therefore, of half the footprint of the ground (60ξ120 mm^2^). Other parameters of Antenna I were theoretically calculated based on our previous work [14] and optimized through simulations as: *l* = 2*w* = 88.4 mm, *x_f_* = 23 mm.

Figure 2 shows the simulated reflection coefficient curve of Antenna I. The resonance frequency of the ${\mathrm{TM}}_{1,1,0}^{\mathrm{HM}}$ mode (E-field distribution as shown in the inset) was 2.39 GHz, and the corresponding −10 dB impedance band covered 2.35~2.43 GHz. The obtained simulation results show that Antenna I can cover the MBAN band (2.36~2.4 GHz) but fails to provide a full coverage over the 2.45 GHz ISM band (2.4~2.4835 GHz).

## 3. Textile Bandwidth-Enhanced Slotted HMSIC Antenna

Inspired by our previous work [26], a straight slot perpendicular to the radiation aperture was cut out from the patch to excite two close resonances for bandwidth enhancement. Figure 3 shows the geometry of the textile slotted HMSIC antenna (Antenna II). Figure 4 shows its simulated return loss curves with multiple values of *l_s_*, *w_s_* and *y_s_*. When *l_s_* increased, the lower resonance frequency did not change, while the higher resonance frequency increased. When *w_s_* increased, the higher resonance frequency stayed unchanged, but the lower resonance frequency increased. Additionally, when *y_s_* changed, the impedance matching could be changed. Based on parameter sweeps, the dimensions of Antenna II were optimized as: *l* = 2*w* = 88.4 mm, *x_f_* = 23 mm, *y_f_* = = 25 mm, *l_s_* = 28.5 mm, *w_s_* = 6.5 mm, *y_s_* = 1 mm.

Figure 5a,b shows the simulated curves of the reflection coefficient and AR of Antenna II, respectively. At 2.33 and 2.45 GHz, two clear and deep notches are observed in Figure 5a. The |S_11_| parameter was below −10 dB in 2.29~2.50 GHz (9.0% fractional bandwidth), and both the MBAN and 2.45 GHz ISM bands were covered. The lowest AR was observed at 2.42 GHz with its simulated value being 3.2 dB, which indicates the existence of CP radiation. When the frequency was within the range of 2.29~2.34 GHz, the simulated AR was over 10 dB, and Antenna II could be mainly predicted to be in the LP mode. To conclude, the polarization mode of Antenna II was circular at 2.42 GHz and linear at a lower frequency between 2.29 and 2.34 GHz.

## 4. Textile Polarization-Reconfigurable HMSIC Antenna Based on Snap Buttons

### 4.1. Geometry Design

Reference [27] illustrated that adding shorting vias at the radiation apertures of a coupled-mode substrate-integrated cavity (CMSIC) antenna leads to a shift of the −10 dB impedance band. Considering that a CMSIC antenna consists of two coupled back-to-back HMSIC antennas, it was, therefore, assumed that adding switches at the radiation aperture of Antenna III also leads to a shift of the band and an increase in the AR of 2.42 GHz. Based on that, two sets of copper snap buttons were installed along the patch edge as switches to shift the band. Each set of snap buttons consisted of a male button fixed on the ground and a removable female button. Two holes were also cut out from the patch above the male buttons; therefore, the direct shorting between the patch and the male buttons was avoided. Figure 6 shows the geometry of the textile slotted HMSIC antenna with snap buttons (Antenna III), and the OFF and ON states of the snap buttons are shown in the insets.

Parameters *l*, *w*, *x_f_*, *y_f_*, *l_s_*, *w_s_*, *y_s_* for Antenna III were of the same values of those for Antenna II. The reflection coefficient of Antenna III was simulated with different values of *y*_*v*1_ and *d_yv_*, and the simulated curves are presented in Figure 7. The simulated button-OFF curve barely changed with different *y*_*v*1_ and *d_yv_*. For the button-ON scenario, however, the simulated curve shifted left when *y*_*v*1_ or *d_yv_* increased, and changing *y*_*v*1_ did not lead to the significant change of the impedance matching while changing *d_yv_* did. Through parameter sweeps, the values of *y*_*v*1_ and *d_yv_* were, respectively, optimized to be 38.4 mm and 1 mm to yield a maximum total frequency coverage with buttons OFF and ON and to keep the |S_11_| parameter below −10 dB at the frequency where the lowest AR was obtained.

### 4.2. Simulation Results

As shown in Figure 8, the simulated curves of the reflection coefficient and AR of Antenna III were shifted right when the buttons are switched from OFF to ON. When the buttons were OFF, Antenna III showed the same resonance frequencies (2.33 and 2.45 GHz), −10 dB impedance band (2.29~2.50 GHz) compared to Antenna II, and the lowest simulated AR which is still obtained at 2.42 GHz was 3.2 dB. When the buttons were ON, the two resonance frequencies and the −10 dB band became 2.45 GHz, 2.57 GHz and 2.41~2.62 GHz, respectively, and the AR at 2.42 GHz increased to 13.1 dB.

Figure 9 shows the simulated vector electric field (E-field) inside the half cavity of Antenna III at 2.42 GHz. With buttons OFF, the ${\mathrm{TM}}_{1,1,0}^{\mathrm{QM}}$ mode was excited within the right quarter cavity, and the resonance was coupled into the left counterpart with an approximate 90° phase delay. According to the simulated fringing fields, the antenna can be considered as two orthogonal magnetic currents (red dashed lines with arrows). When the phase increased, a clockwise rotation was observed for the composed vector magnetic current ***M*** (red solid lines with arrows), and, therefore, indicates the right-hand circular polarization (RHCP) radiation with buttons OFF. With buttons ON, the distribution of the E-field accords with the ${\mathrm{TM}}_{1,1,0}^{\mathrm{HM}}$ mode, and the antenna acts as a horizontal magnetic dipole (red dashed lines with arrows) with a LP radiation.

The free-space gain patterns of Antenna III were simulated at 2.42 GHz and shown in Figure 10. When buttons were OFF, the maximum ${\mathrm{Gain}}_{\mathrm{RHCP}}$ was 6.1 dBc, and the maximum ${\mathrm{Gain}}_{\mathrm{RHCP}}$ was −8.6 dBc lower than the former. When the buttons were ON, the coplanar LP gains in both the *xz* and *yz* planes were of a main beam towards the +*z* direction with a large beam width. The maximum coplanar LP gain was 6.6 dBi, and the maximum cross LP gain was 13.3 dBi lower than the former. Therefore, the pattern simulation results further indicate the RHCP and LP radiations of Antenna III at 2.42 GHz with buttons OFF and ON, respectively. In addition, the simulated radiation efficiency of Antenna III in free space was 95% with buttons OFF and 97% with buttons ON.

A 300 × 300 × 60 mm^3^ skin-fat-muscle phantom, as shown in Figure 11, was applied to study the antenna performance when Antenna III was practically worn on the human body. The phantom was located under Antenna III, and there was a 1 mm gap between Antenna III and the phantom. The densities and electrical characteristics at 2.42 GHz for each tissue layers were selected based on [28], as shown in Table 1. Figure 8a illustrates that the simulated return loss curves of Antenna III on phantom well agree with those in free space, and thus, indicates the robust frequency characteristics of Antenna III against the influence from the human tissues.

The simulated patterns of the CP and LP gains of Antenna III on the phantom at 2.42 GHz are shown in Figure 12. When buttons were OFF, the maximum ${\mathrm{Gain}}_{\mathrm{RHCP}}$ and ${\mathrm{Gain}}_{\mathrm{LHCP}}$ were 4.9 and 0.2 dBc, respectively. When buttons were ON, the maximum coplanar-polarized gain was 6.4 dBi, and the maximum cross-polarized gain was −5.9 dBi. Compared to the free-space patterns in Figure 10, the backward radiations were reduced because of the absorption and reflection effects of human tissues in the phantom. Additionally, human tissues as lossy dielectric materials can absorb a certain proportion of the antenna radiation. Simulation results on phantom showed that the radiation efficiency reduced to 85% with buttons OFF and 89% with buttons ON at 2.42 GHz. The central position of the radiation aperture and the sufficient electromagnetic isolation provided by the present ground lead to a rather small efficiency loss, which is favorable in wearable applications.

With an input power of 0.5 W applied into the feeding port, the SAR (1 g average) in the phantom at 2.42 GHz was simulated for Antenna III. The simulated SAR distributions with a 0.5 W input power are presented in Figure 13, and the simulated maximum SARs in the phantom with buttons OFF and ON were, respectively, 0.53 and 0.32 W/kg which satisfied the IEEE (≤1.6 W/kg) and EN (≤2.0 W/kg) safety requirements. Therefore, the present ground provides an enough electromagnetic shielding for the wearer, and the radiation of Antenna III does not lead to electromagnetic damage in the human tissues.

### 4.3. Measurement Results

A prototype of is fabricated for Antenna III using computerized embroidery. As shown in Figure 14, the employed materials for Antenna III include PF-4 foam (Cuming Microwave) as substrate, conductive fabric NCS95R-CR as patch and ground, conductive epoxy CW2460 (CircuitWorks) for the connection between the SMA connector and conductive fabric and conductive thread Shieldit 117/17 2PLY linearly embroidered as sidewall. Three passes of embroidery were applied for the sidewall to lower the loss for a higher radiation efficiency, and the distance between two adjacent stitches was set to be 1.5 mm.

The reflection coefficient curves of Antenna III were measured by a vector network analyzer and presented in Figure 15. In free space, the measured curve showed a reasonable agreement with the simulation results. According to the measured results, Antenna III resonated at 2.33 and 2.45 GHz with buttons OFF, and the −10 dB impedance band covered 2.29~2.50 GHz with a 9.1% fractional bandwidth. When the buttons were ON, the measured resonance frequencies, respectively, shifted to 2.46 and 2.58 GHz, and the band shifted to 2.42~2.63 GHz (8.3% fractional bandwidth). Therefore, the band of Antenna III can be mechanically reconfigured to cover 2.29~2.63 GHz with a 13.9% fractional bandwidth. When the antenna was practically on the body of a volunteer, the measurement results showed that the reflection coefficient curves almost overlap with those measured in free space as expected from simulations. Additionally, Antenna III was measured on a cylindrical PVC frame which bends around the *x*-axis or *y*-axis with an 8 cm bending radius. In such bending scenarios, both resonance frequencies with buttons OFF slightly increased, but the measured −10 impedance band still well covered the target bands. The observed frequency shift can be due to the decrease in cavity width/length caused by physical deformations of the flexible fabric and foam substrate during bending.

The 2.42 GHz gain patterns of Antenna III in free space were practically measured in an anechoic chamber and are presented in Figure 16. All measured gain patterns well accord with simulation expectations in terms of the pattern shapes. When buttons were OFF, the maximum ${\mathrm{Gain}}_{\mathrm{RHCP}}$ and ${\mathrm{Gain}}_{\mathrm{LHCP}}$ were 5.8 and −1.2 dBc, respectively. When the buttons were ON, the maximum coplanar-polarized gain was 5.1 dBi, and the maximum cross-polarized gain was −2.7 dBi. Due to limited measurement equipment, the SAR, radiation efficiency and on-body gain patterns of the proposed antenna were not practically measured in this work.

## 5. Conclusions

A novel textile bandwidth-enhanced polarization-reconfigurable HMSIC antenna was designed for wearable applications. Firstly, by applying a slot and two sets of metallic snap buttons in the geometry of a textile HMSIC antenna, not only the separate bandwidth with buttons OFF or ON was enhanced, but a larger frequency range can be covered by changing the state of the buttons to reconfigure the band. As shown in Table 2, the low-loss characteristics of the employed antenna materials leads to the less competitive bandwidth with buttons in the OFF or ON state compared with other bandwidth-enhanced wearable/flexible HMSIC antennas, but its reconfigurable characteristic enable its operation in a wide frequency range of 2.29~2.63 GHz (13.9% fractional bandwidth), and enable its advantageous radiation efficiency and cavity size. Secondly, by adjusting the position of the snap buttons, the RHCP and LP radiations were, respectively, realized at 2.42 GHz with buttons OFF and ON. Compared with the DC bias circuits in reference [20,21,22,23], the mechanical switches based on snap buttons further enhanced the compatibility of the proposed antenna with textile materials and require no soldering to install or DC power to function.

## Figures and Tables

**Figure 1 micromachines-14-00934-f001:**
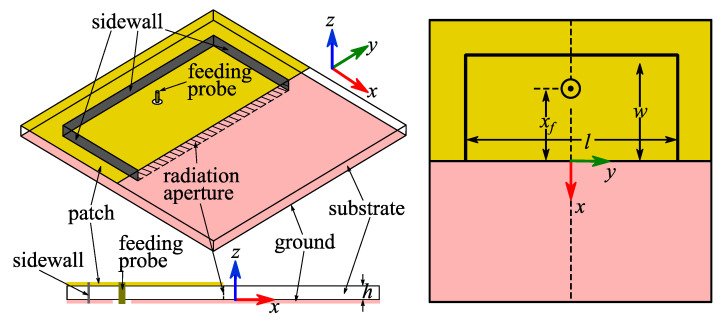
Geometries of Antenna I.

**Figure 2 micromachines-14-00934-f002:**
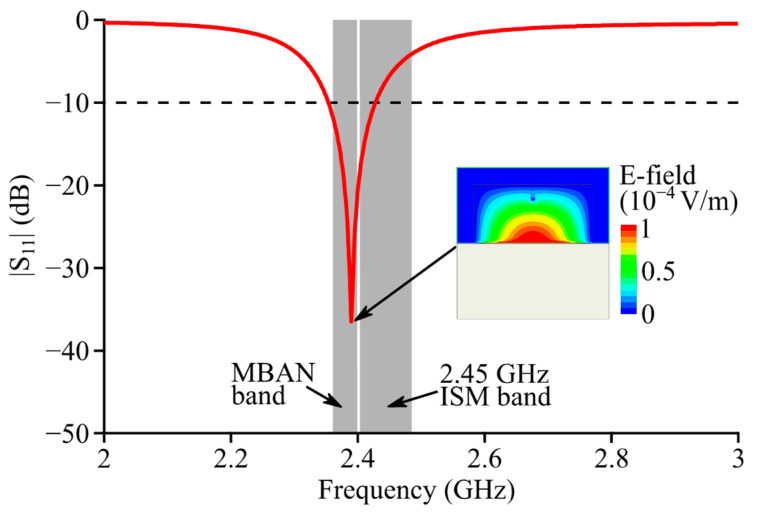
Simulated reflection coefficient curve of Antenna I.

**Figure 3 micromachines-14-00934-f003:**
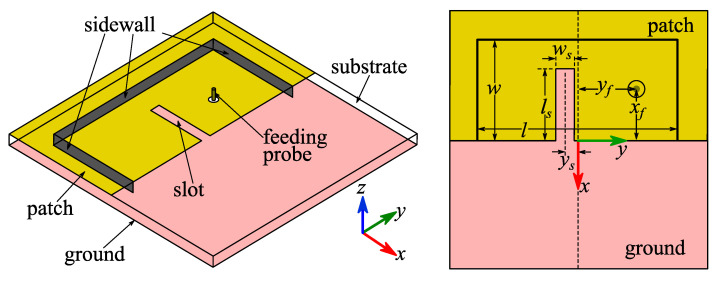
Geometries of Antenna II.

**Figure 4 micromachines-14-00934-f004:**
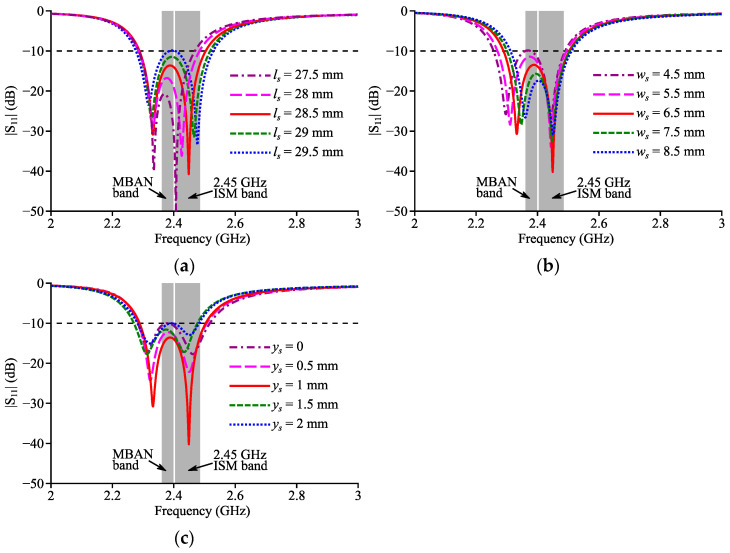
Simulated reflection coefficient curves of Antenna II with different (**a**) *l_s_*, (**b**) *w_s_* and (**c**) *y_s_*.

**Figure 5 micromachines-14-00934-f005:**
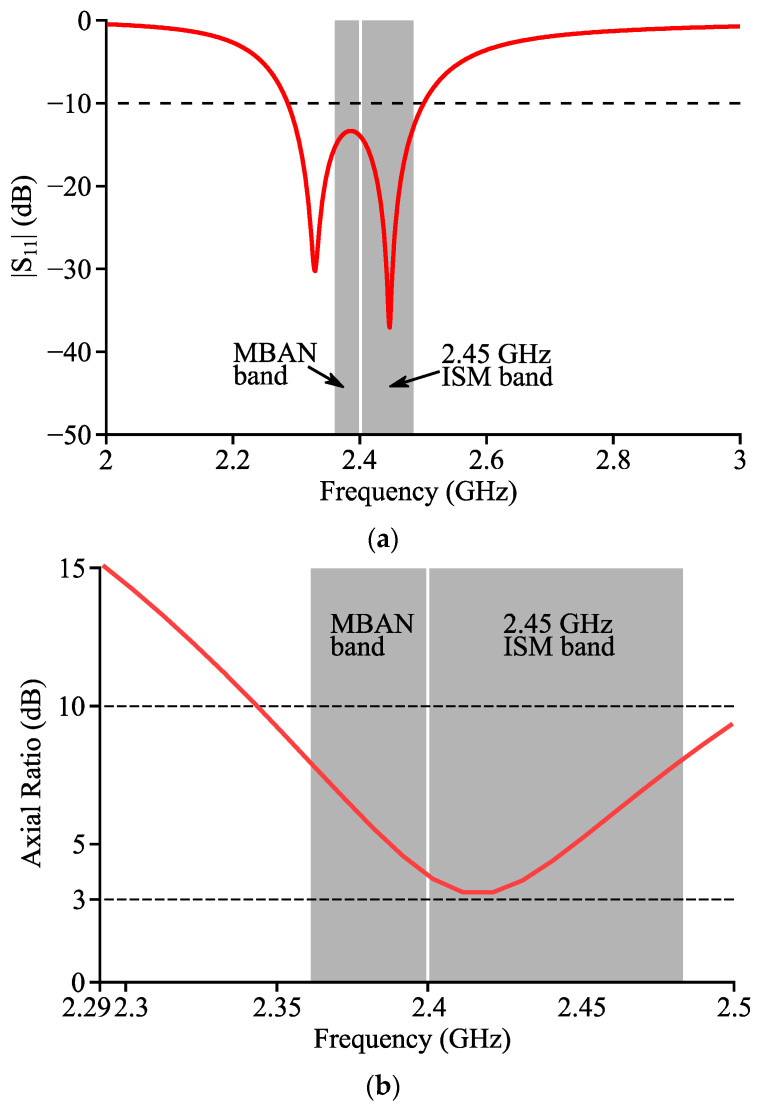
Simulated curves of (**a**) reflection coefficient and (**b**) AR of Antenna II.

**Figure 6 micromachines-14-00934-f006:**
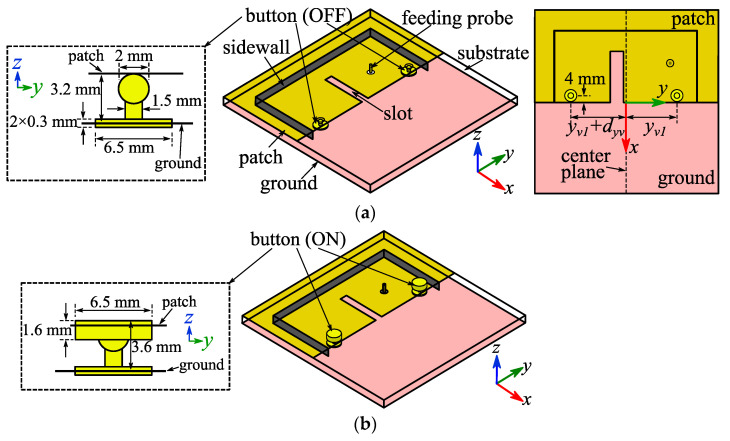
Geometries of Antenna III: (**a**) buttons OFF; (**b**) buttons ON.

**Figure 7 micromachines-14-00934-f007:**
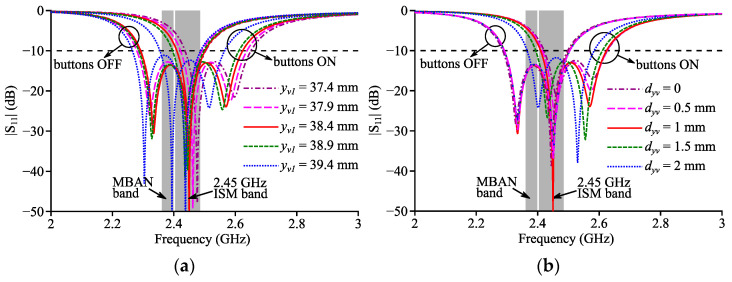
Simulated reflection coefficient curves of Antenna III with different (**a**) *y*_*v*1_ and (**b**) *d_yv_*.

**Figure 8 micromachines-14-00934-f008:**
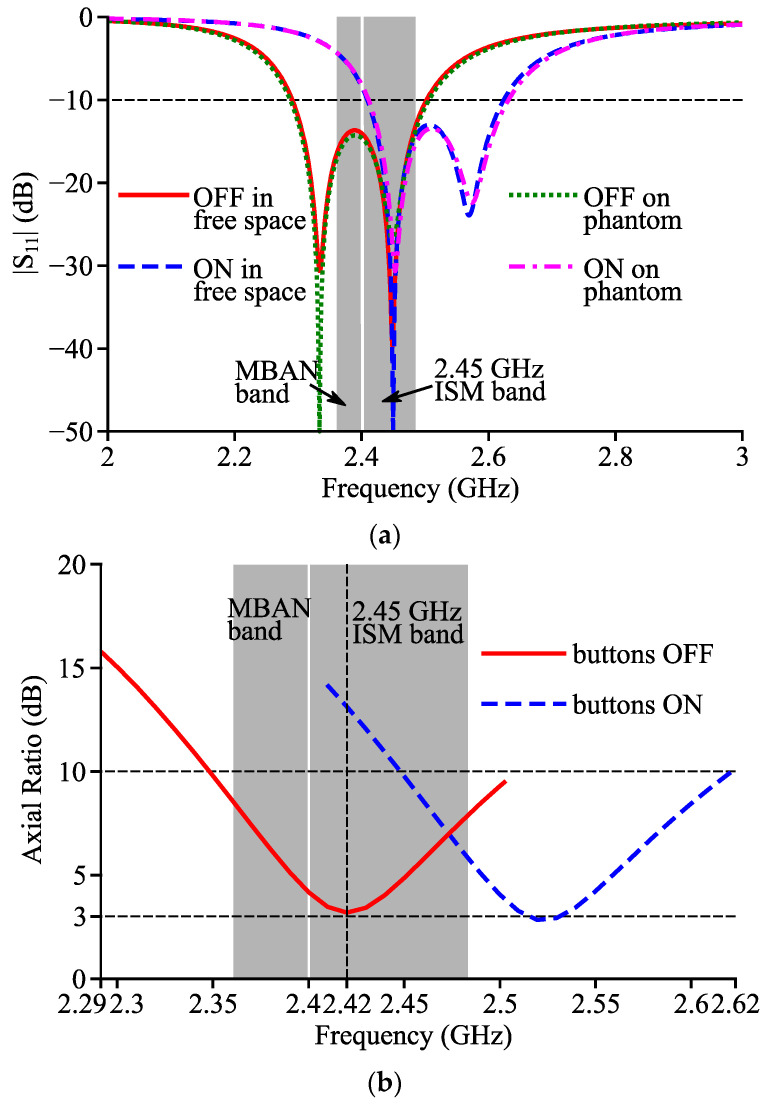
Simulated (**a**) |S_11_| curves and (**b**) AR curves of Antenna III.

**Figure 9 micromachines-14-00934-f009:**
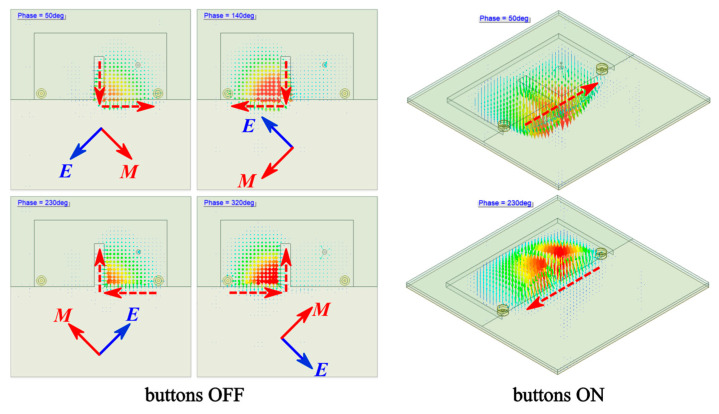
Simulated distributions of internal vector electric field of Antenna III at 2.42 GHz.

**Figure 10 micromachines-14-00934-f010:**
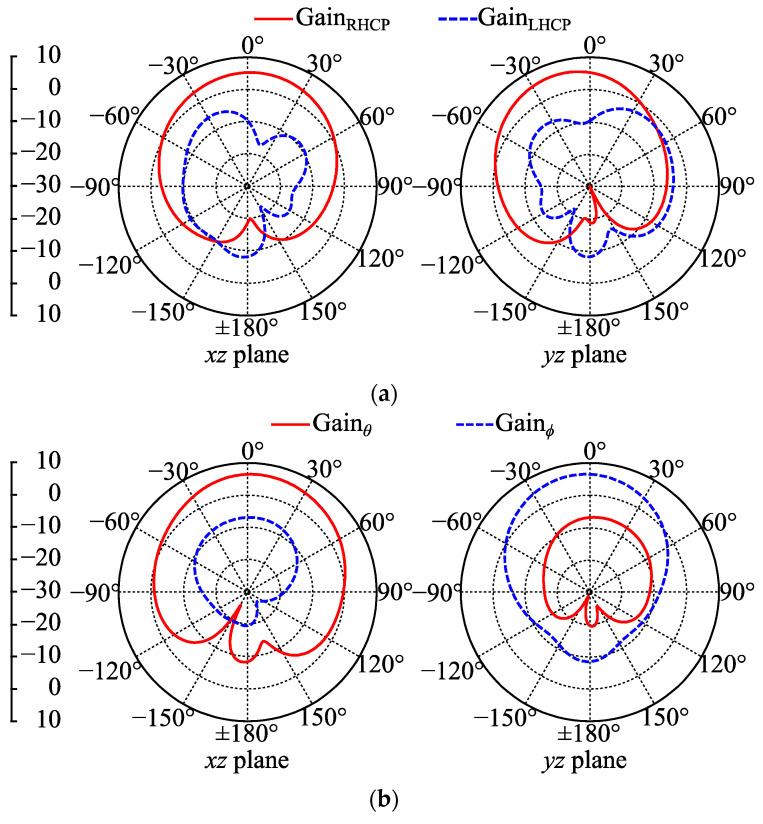
Simulated free-space gain patterns of Antenna III at 2.42 GHz: (**a**) buttons OFF; (**b**) buttons ON.

**Figure 11 micromachines-14-00934-f011:**
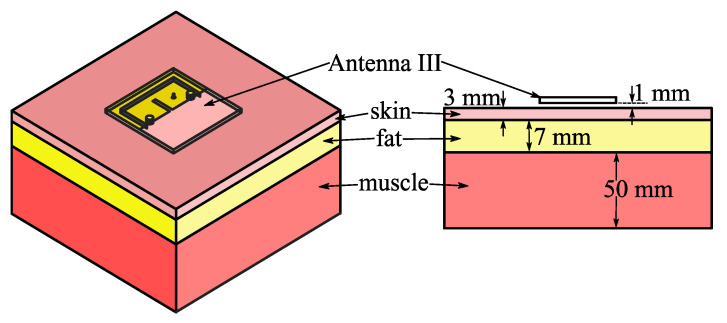
Geometry of skin-fat-muscle phantom.

**Figure 12 micromachines-14-00934-f012:**
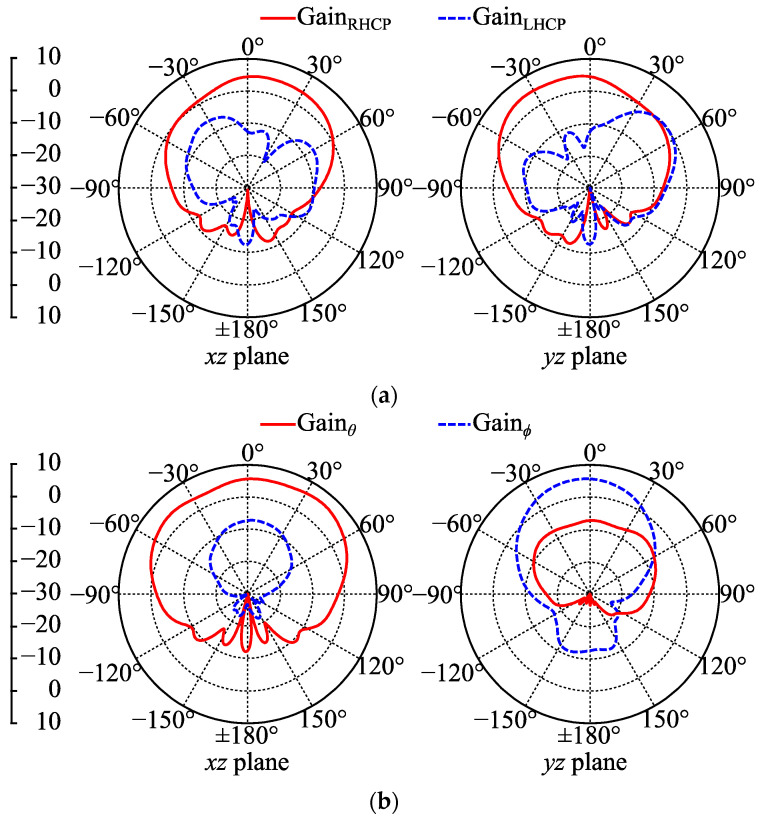
Simulated gain patterns of Antenna III on phantom at 2.42 GHz: (**a**) buttons OFF; (**b**) buttons ON.

**Figure 13 micromachines-14-00934-f013:**
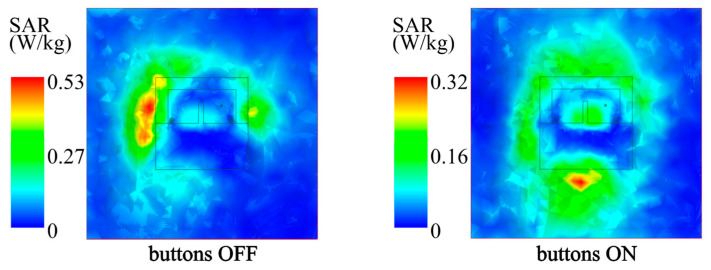
Simulated SAR of Antenna III in phantom at 2.42 GHz.

**Figure 14 micromachines-14-00934-f014:**
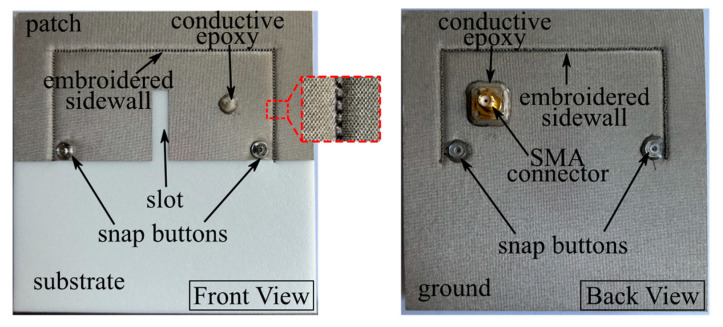
Fabricated prototype of Antenna III.

**Figure 15 micromachines-14-00934-f015:**
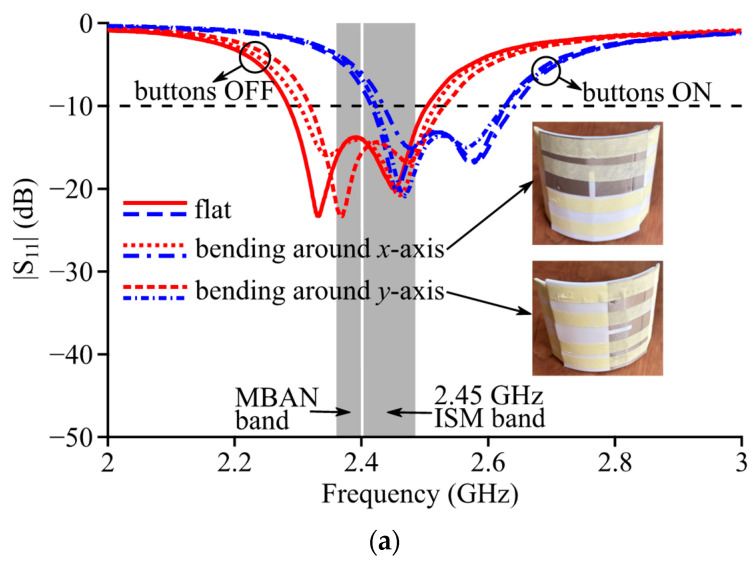
Measured reflection coefficient curves of Antenna III: (**a**) in free space; (**b**) on human body.

**Figure 16 micromachines-14-00934-f016:**
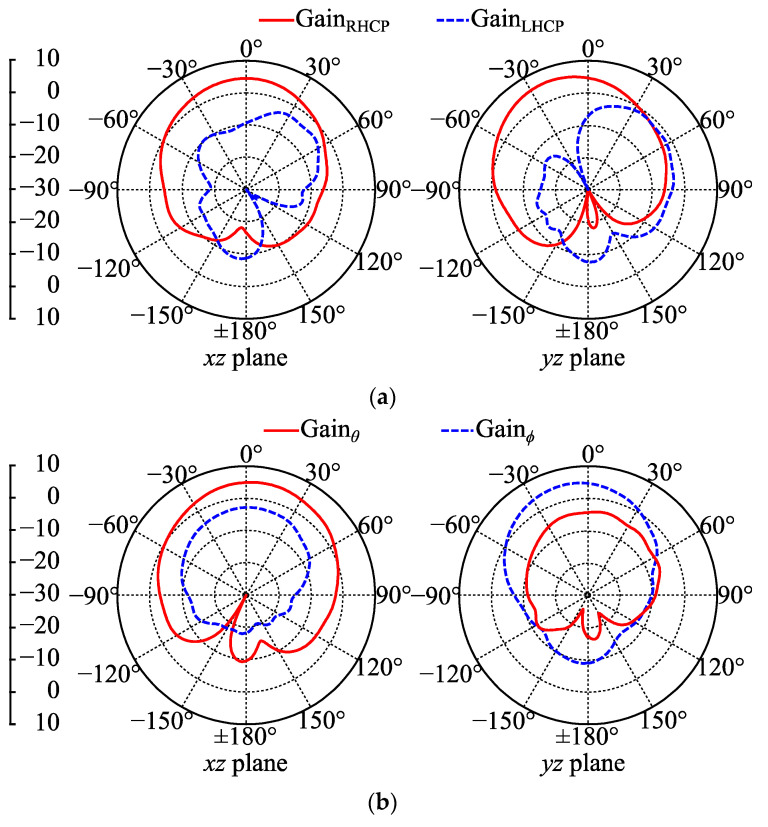
Measured free-space gain patterns of Antenna III at 2.42 GHz: (**a**) buttons OFF; (**b**) buttons ON.

**Table 1 micromachines-14-00934-t001:** Densities and Electrical Characteristics of Each Layers of Phantom.

Layer	Density (kg/m^3^)	Relative Permittivity	Conductivity (S/m)
Skin	1100	38.09	1.43
Fat	910	5.29	0.1
Muscle	1041	52.8	1.69

**Table 2 micromachines-14-00934-t002:** Comparison with Other Wearable/Flexible Bandwidth-Enhanced HMSIC Antennas.

Ref.	Freq. (GHz)	Bandwidth (%)	Efficiency (%)	Cavity Size (λ_ε_^2^)	Antenna Size (λ_ε_^2^)	Polarization Reconfiguration
[15]	5.5	23.7	85	0.364	Not given	No
[16]	2.45	6	Not given	0.22	0.673	No
[18]	5.5	14.7	95	1.073	3.527	No
[19]	5.5	15.7	94	0.245	1.754	No
This work	2.4 (OFF)	9.1 (OFF)	95 (OFF)	0.266 (OFF)	0.980 (OFF)	Yes (2.42 GHz)
	2.52 (ON)	8.3 (ON)	97 (ON)	0.285 (ON)	1.05 (ON)	

## Data Availability

All data are included within the manuscript.

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
