# Peer review of "Textile Bandwidth-Enhanced Polarization-Reconfigurable Half-Mode Substrate-Integrated Cavity Antenna"

_micromachines, 2023, doi:10.3390/mi14050934_

Round 1

Reviewer 1 Report

The paper describes a texile antenna with a rather large bandwidth and reconfigurability. A reasonable description of the design approach is present and the simulations are supported by the measurement. After some minor corrections, it can be published.

The details of the points which needs corrections follow.

The author's language is rather clear, but several English words are incorrect or misspelled. Among them, I must quote the use of "dynamic", widespread in the whole manuscript starting from the abstract, which is incorrect. The meaning of "dynamic" is "behavior which can be modified in real-time". An antenna with some PIN or varactor diodes can "dynamically modify its behavior". An antenna, like the one described here, with snap buttons which must be act by the user's hands DOES NOT "dynamically modify its behavior", but is a "reconfigurable" antenna. This use of "dynamical" must be correct, especially in the abstract, as it can be misleading for a reader. Other English misuse are "practically" in row 74 (which can simply be dropped) or "for" for "toward" in row 86, but the list is by no means exaustive.

row 36: please, insert the meaning of HMSIC also here. Acronyms should be avoided in the Abstract, unless, as in the present case, they are used again in the abstract. But their meaning must be repeated in the body of the paper when tey are used for the first time. The same happens for "CP/LP", row 25, which is an useless acronym here, and should be introduced in the body of the manuscript.

row 80: it is not clear whether ground and patch have the same size or not. It seems not, but this is unclear in the text, and should be clarified.

Fig. 5b: please, add MBAN and ISM also here (like in Fig. 5a), and also horizontal dashed lines to mark 3 dB and the axial ratio over which a good LP is obtained (probably 10 dB). It seems that the polarization is neither CP nor LP in the MBAN. Please, comment on this.

row 143: where is dyv  defined?

Fig. 8a: please, divide into two figures (one for free space and the other for phantom) as now the curves are hardly distinguishable.

Fig. 8b: same as Fig. 5b

row 178: instead of the LHCP gain, insert the X-pol level w.r.t the Co-pol, as this is a more interesting parameter (the same for the LP case in row 181)

row 210: the reduction in efficiency is rather small, and this is a clear pro for this antenna. I imagine that this depends on the position of the radiation aperture, but a remark on this small efficiency loss, and a comment on the reason why, is important.

row 217: why 0.5 W? For this power, SAR limits are fulfilled, but for a larger one they are not. So, the statement of row 222 must refer to the power used in simulations.

Fig. 15a: please, divide into two figures (for the same reason of Fog. 8a).

Table 2: In the comparison, it is important to compare the total antenna size (instead, or in addition, of the cavity size). Please, add this info. (in the last row of this table there is a typo "OFF" for "ON")

ref [27]: tissue data should be referenced from Gabriel papers [A], or maybe from a book on wearable antennas ([B] or [C]), not from a paper by the same authors (which describe an antenna, not tissue data).

[A] Gabriel, C. (1996) Compilation of the dielectric properties of body tissues at RF and microwave frequencies.
Brooks Air Force Technical Report, AL/OE-TR-1996-0037.

[B] Hall P.S., Hao Y. - Antennas And Propagation for Body-Centric Wireless Communications-Artech House  2nd ed (2012)  

[C] Wang J.,Wang Q. - Body Area Communications_ Channel Modeling, Communication Systems, and EMC (2013, Wiley-IEEE Press)

See "Comments and Suggestions for Authors
"

Reviewer 2 Report

The authors have proposed a textile bandwidth-enhanced polarization reconfigurable HMSIC antenna. Some of my comments are as follows:

1) Polarization reconfigurable operation is not clear. Please discuss how the authors are getting polarization reconfigurable operation.

2) Please discuss the losses for the half-mode structure.

3) This type of antenna is very common in the literature.  It is advised to discuss the novelty of the proposed antenna clearly.

4) In the introduction section please add the state of the art of this type of antenna so that the readers can get an overall idea of this type of antenna.

There are some grammatical mistakes. Please correct all those mistakes. 

Round 2

Reviewer 2 Report

Thanks for addressing my comments. I have no more comments. Please improve the quality of English for this paper.  

The quality of the English in this paper is not good. It is advised to edit the language so that the readers of this journal can understand it clearly.